# Are TikTok Algorithms Influencing Users' Self-Perceived Identities and Personal Values? A Mini Review

**Claudiu Gabriel Ionescu** *,† and **Monica Licu** †

Medical Ethics Department, Carol Davila University of Medicine and Pharmacy, 050474 Bucharest, Romania; monica.licu@umfcd.ro

**\*** Correspondence: claudiu.ionescu@drd.umfcd.ro

**†** These authors contributed equally to this work.

**Abstract:** The use of TikTok is more widespread now than ever, and it has a big impact on users' daily lives, with self-perceived identity and personal values being topics of interest in light of the algorithmically curated content. This mini-review summarizes current findings related to the TikTok algorithm, and the impact it has on self-perceived identity, personal values, or related concepts of the Self. We pass through the contents of algorithmic literacy and emphasize its importance along with users' attitudes toward algorithms. In the first part of our results, we show conceptual models of algorithms like the crystal framework, platform spirit, and collective imaginaries. In the second part, we talk about the degree of impact a social media algorithm may exert over an individual's sense of self, understanding how the algorithmized self and domesticated algorithm are trying to sum up the dual development of this relationship. In the end, with the concept of Personal Engagement and the role of cognitive biases, we summarize the current findings and discuss the questions that still need to be addressed. Performing research on the topic of social media, especially TikTok, poses ethical, cultural, and regulatory challenges for researchers. That is why we will discuss the main theoretical frameworks that were published with their attached current studies and their impact on the current theoretical models as well as the limitations within these studies. Finally, we discuss further topics of interest related to the subject and possible perspectives, as well as recommendations regarding future research in areas like impact on personal values and identity, cognitive biases, and algorithmic literacy.

**Keywords:** social media; TikTok; personal values; self; identity; algorithms

## 1. Introduction

Many individuals use social media, especially TikTok, with many of them scoring highly on various scales of social media use. With the platform being declared the most addictive one (Smith and Short 2022), TikTok is also the most used social media platform by generation Z in the world, with over 1 billion users monthly (Gu et al. 2022; Boeker and Urman 2022). Its implications for human behavior are not completely understood yet (Smith and Short 2022). It is already shown that a TikTok personal experience is shaped by the individuals' motivations and behaviors which, in the perspective of values-aligned and personalized content bought by the algorithm, can enhance positive experiences and promote psychological wellbeing (Ostic et al. 2021; Naslund et al. 2020) while in the same time can lead through cognitive biases which are flaws or distortions in judgment and decision making that can lead to poor outcomes, to the "echo-chamber" effect to isolation, rigid policy attitudes or radicalization in various aspects of personal values and beliefs with a major potential societal impact not yet fully addressed (Binder and Kenyon 2022; Cinelli et al. 2021; Boucher et al. 2021; Sasahara et al. 2021).

The role of human self-perceived identity and personal values has been a prominent area of theorization in relation to social media algorithms. Self-perceived identity refers

to an individual's subjective interpretation and understanding of themselves, their environment, and various aspects of their lives and encompasses cognitive, emotional, and behavioral consistency in how they perceive and define themselves (Ersanlı and Şanlı 2015). Little academic analysis tried to deepen the knowledge of this topic throughout time, thus becoming overly simplified (Boyd et al. 2021; Silva et al. 2020; Hynes and Wilson 2016; Chen et al. 2014). Human–algorithm interaction has emerged as a pressing area in social media research, probably because algorithms curate and govern most forms of communication on these platforms. Social media algorithms are computational models for transforming data into personalized content that populates a user's social feeds, such as TikTok's "ForYouPage", but may be influenced by users' perceptions (Bucher 2020). TikTok has an extremely advanced and sophisticated algorithm system, especially in terms of participation, content, and types of interaction, which makes the addiction problem more severe than on all the other popular social media platforms, especially among younger generations (Smith and Short 2022). Due to the opaqueness or "blackbox-ed" nature of algorithms, users experience them just through their perceptions. Because what people see on social media is largely personalized, it shapes how people see themselves and others (Bhandari and Bimo 2022) but also impacts their behavior on social media platforms (DeVito 2021). Research on human–algorithm interaction involves investigations along multiple fronts: how social media algorithms function, how individuals perceive them, and the effects of social media algorithms on the Self. Firstly, research focused on theories on how social media algorithms operate and their relationship with marginalized identities (DeVito 2021; Simpson and Semaan 2021). This area of research suggests that identity maintenance and development are performed through the curated feed of the algorithm. It is important to note that individual differences and personal circumstances also play a significant role in developing personal values and attitudes within social media usage and may develop a sense of authenticity and coherence in one's self-perceived identity.

With algorithms mediating communications and changing relationships and society, we support the urgent need to better understand and assess the current knowledge on its impact not only on individual personal values and identities but also on public attitudes, polarization, developer transparency, and user privacy, and give cohesive points of discussion for the future. Effects on individuals will remain a topic of special interest, with emergent questions regarding algorithm access, transparency, and accountability waiting for answers to be given. Our study aimed to present the current state of research, give a broad overview of algorithm concepts and their impact, address questions that still needed to be answered, and highlight perspectives on future studies regarding both psychology, social ethics, and artificial intelligence research.

## 2. Materials and Methods

Studies included in the review consisted of published English-language peer-reviewed articles focused on examining social media algorithms and their relationship with the Self, including our variables of interest: self-perceived identity and personal values. We utilized the following databases: Google Scholar, Elicit, PsycINFO, PubMed, and Science Direct conducted electronic searches from March 2023 to April 2023, examining articles published at any time prior to 25 March 2023. Searches were conducted with terms related to personal values (i.e., "personal values", "moral values", "self", "ethics", "moral attitudes", or "ethical values") and TikTok (i.e., "social media", "TikTok", "Facebook", "Instagram", "algorithm", or "social networking sites"). The concepts of "self-perceived identity" and "personal values" were very scarcely mentioned throughout the literature, thus we included broader concepts like "self-representation", "self-image", "self-making" to attain a broader perspective on our primary objectives. All published articles found using the above search terms and deemed to be related to the topic of focus were included based on their relevance.

### 3. How Users Understand and Relate to the Algorithm Has an Impact by Itself

*3.1. The Role of Algorithm Awareness among Users*

The algorithm awareness among users is of major importance in the overall questioning of the topic. It is shown that the degree of algorithmic literacy impacts the overall experience and influences the personal social media experience (Taylor and Choi 2022). Due to the opaque nature of algorithms, users experience them through their perceptions as they interact with them. A study focused its research on how the users perceive the interaction with the algorithm, following the hypothesis that users make judgments about how responsive and insensitive the algorithmically curated content is and how well the algorithm "listens" to information. In this study, all the participants were, of course, algorithm-aware. As a result, the novel concepts of PAR (perceived algorithm responsiveness) and PAI (perceived algorithm insensitivity) were only weakly correlated to algorithmic literacy, and PAR was a significant predictor of people's social media enjoyment (Taylor and Choi 2022). Another study showed that only 61% of users were aware or had low awareness of the algorithm, with a cluster analysis creating an algorithm-awareness typology: the unaware, the uncertain, the affirmative, the neutral, the sceptic, and the critical (Gran et al. 2021). Also, the higher the levels of education, the more negative the attitudes towards algorithms are, with the digital divide being exacerbated by smart machine learning infrastructure. This may have an important impact on informed decision-making processes in the context of data-driven arrangements and social injustice (Dencik et al. 2016, 2018, 2019). However, algorithmic literacy was studied by identifying context-specific sense-making strategies of algorithms, expectancy violations, and explicit personalization cues as users had intuitive and experience-based insights into feed personalization that did not automatically enable them to verbalize these (Swart et al. 2018; Swart 2021). Also, personality factors and other demographic variables that may impact the degree of awareness, still need to be studied (Lutz and Hoffmann 2019; Dutton and Graham 2019).

*3.2. From Folk Theories to Personal Engagement and Identity Strainer*

Folk Theories "are intuitive, informal theories that individuals develop to explain the outcomes, effects, or consequences of technological systems, which guide reactions to and behavior towards said systems" (DeVito 2021), with one of the most common being *Personal Engagement Theory*. It means social media feeds are curated through personal engagement metrics, which are based on digital traces, including what content the user previously liked, commented on, and viewed the most (Eslami et al. 2016). Another theory worth mentioning, the *Identity Strainer Theory* highlighted that the curation happens as an identity strainer, which means that the algorithm curates the social feed by suppressing content related to marginalized social identities based on race and ethnicity, body size, and physical appearance, LGBTQ identity, and political and social justice group affiliation (Karizat et al. 2021). In reverse, the concepts of *algorithmic privilege* and *algorithmic representational harm* refer to the harm users experience when they lack algorithmic privilege and are subjected to algorithmic symbolic annihilation, leading them to shape their algorithmic identities, thus aligning with how they understand themselves, as well as resisting the suppression of marginalized social identities (Karizat et al. 2021). In the same study, the interaction is viewed as a co-production or a bidirectional relationship, which in turn can be treated as input data for the algorithmic decisions themselves. These findings were similar to other studies concerning LGBTQ users (DeVito 2022; Simpson and Semaan 2021).

*3.3. The Concept of the Crystal Framework*

"The Crystal Framework" is a conceptual model metaphor of crystals and their properties put into an analytic frame that includes the reflection of self-concepts that are both multifaceted and multidirectional, shaping perspectives on others by orienting them to recognize parts of themselves refracted in others and to experience ephemeral, diffracted connections with various groups as a virtual community. This qualitative study identified more positive attitudes towards algorithms, especially TikTok's' (Lee et al. 2022). The main

characteristics of the "crystal" were: reflective (parts of their self-concept were reflected back to them in the feed); multifaceted; has a refinement strategy; is diffractive. It is one of the first conceptual models out there to point out that the algorithm is shaping self-concepts without specifically addressing how this may happen. Anywise, an interesting point of the study was that when users view algorithms as exploiting users for monetary gain or manipulating attention, they may not believe the content is an accurate reflection of themselves, which may be changing and aligning more with the "crystal". Causes of this may be internal or external; the latter resulted from a "broken crystal" translated into the influences companies, financial interests, politics, and even TikTok policy may adopt.

### 3.4. Users' Attitudes towards TikTok Algorithms

Regarding users' attitudes towards algorithms, a study identified algorithms as being perceived as confining, practical, reductive, intangible, and exploitative. The authors emphasized "digital irritation" as a central emotional response with a small but significant potential to inspire future political actions against datafication (Ytre-Arne and Moe 2021; Ytre-Arne 2023). In the opposite direction, users describe TikTok as a safe space where users can be themselves, feel included in a community, and engage meaningfully. But, contradictory, the algorithm is perceived as harmful because it tries to manipulate and drive users towards specific videos that increase their "addiction" to the platform. Users consider some of the recommendations on the ForYou Page to be questionable because they aim at persuading or nudging in favor of hashtags and social causes. This contradiction may partly be explained by the fact that participants report their rationalizations in a performative manner to avoid feelings of dissonance while attempting to relate to their own identity (Scalvini 2020).

### 3.5. Additional Concepts: Algorithm Gossip, Collective Algorithmic Imagineries, and Platform Spirit

"Algorithm gossip" addresses folk theories with a focus on content creators who understand that informed theories and strategies pertaining to algorithms can help financial consistency and visibility and, thus, gain more public attention (Bishop 2019). "Collective algorithmic imagineries", or ways of thinking about what algorithms are, what they should be, how they function, and what these imaginations, in turn, make possible, meaning that if users think the algorithm behaves in a particular way according to a specific logic, users tend to adapt their behavior to that belief (Siles et al. 2020; Bucher and Helmond 2018). Also, the algorithm works beyond the interpretative realm of humans in the sense that they are recursively modeled back into algorithmic systems, thus they can influence the decision-making process of the algorithm. Studying this concept, we find that algorithms are perceived as meritocratic, dynamic, and unpredictable, have an experimental attitude with short attention spans, and "selling" the information through emotions rather than facts, all this leading to a "techno-communitarian sentiment" (Øllgard 2023).

In the end, the "Platform Spirit", represented by the constant stream of relatable content, the sense of community among creators and users, and the ability to easily go viral and gain popularity within digital and real life, covers the realm of addiction the algorithm has. Also, adaptation to the spirit and algorithm literacy, even structural theorization, not necessarily structural knowledge of the algorithm, impacts the understanding of the user related to its own decision-making process and the algorithm's as well (DeVito 2021). The response we obtain from this work is that algorithm literacy has a mandatory role in the future influence that constantly changing social media platforms will have on their users.

## 4. Are Self-Perceived Identity and Personal Values Shaped by TikTok Algorithms?

### 4.1. Concept of Self, Algorithmized Identity, and Affective Capitalism

Personal values and self-perceived identity were represented in some studies under the broader relevant concepts of "self-representation", "self-identification", "self-making", or "networked self", referring to the act of identifying oneself as a particular kind of

person (Papacharissi 2011; Strimbu and O'Connell 2019; Tiidenberg and Whelan 2017; Bamberg 2011; Thumim 2012). That is why we have chosen to use these concepts when referring to our studied variables. Another school of thought was "self-symbolizing", with a study showing that those who publicly presented themselves on ephemeral social media internalized their portrayed personality and development over time (Choi et al. 2020). But how in-depth can this process affect self-perception of identity and personal values of users? A study shows the concept of "affective capitalism", where all the desires, emotions, and forms of expressivity are becoming raw materials in a wider economic infrastructure (Hearn 2017, 2019). Through aggregation, abstraction, and categorization, they become consumer profiles, which are ultimately projected back onto them, leading to an "algorithmized identity", which is "an identity formation that works through mathematical inferences on otherwise anonymous beings" (Cheney-Lippold 2011). The challenge arising from this is considering shaping: through what kind of studies can we assess the "Algorithmically Shifted Self"?

### 4.2. Is the Self Algorithmized or the Algorithm Domesticated?

Users may vary in the extent to which they feel that the algorithm is reflective of who they truly are. Then, they may employ a variety of strategies to try to bring the algorithm's recommendations into line with who they are or want to be. (Lee et al. 2022). One study brought the concept of "Algorithmized Self" as a definition for TikTok, as an extension and complication of the previously discussed "Networked Self"; while the latter posits that the self is created through the "reflexive process of fluid associations with social circles", the former understands the self as deriving primarily from a reflexive engagement with previous self-representations rather than with one's social connections. The same study argues that users interact most with aspects of their own personas (Bhandari and Bimo 2022). In contrast, there were described processes as "depuration", in which users consciously train the algorithm to remove content from their feeds (González-Anta et al. 2021), or organic and strategic refinement (Lee et al. 2022). Moreover, as an extended view of the "Networked Self", the *Extended Mind Theory* (Clark and Chalmers 1998) envisions that the environment plays an active role in how cognitive processes are driven. In this regard, non-biological systems such as algorithms may be considered a part of a cognitive process in which our brain performs some operations while the algorithm also plays a causal role in governing an individual's personal values and behavior, as an interactive system or cognitive integration (Chakravarty 2021; Menary 2010). In the light of another theory, "taken as networked and dynamic ecologies that form parts of minds, the environment and all the things in it may together function as a collection of interconnected cognitive systems. This is not only illustrated through human examples but also reinforced by behavior-based artificial intelligence" (Crippen and Rolla 2022), like social media algorithms. The same authors highlight that situations are primary as they may change what we see on faces, but that such cases are not merely psychological projections, that is, consequences of how we represent things in our heads (Crippen 2022). The remaining question is: how will this extended socio-technological self frame situations and make decisions in our best interests regarding our own personal values and behaviors?

Users responded that they were never fully able to control their digital selves and thus integrate it into their routine lives as the TikTok algorithm was constantly misaligned with their personal moral economy (Simpson and Semaan 2021). In conclusion, there is a constant feedback loop between the two concepts, but the degree to which one primarily impacts the other has yet to be studied.

### 4.3. How May the Algorithm Influence Users' Self-Perceived Identity and Personal Values?
4.3.1. Personal Engagement and the Role of Cognitive Biases

The TikTok algorithm does not have direct knowledge of a user's personal values, but it does make assumptions based on users' behavior and interactions. For example, following specific creators was the strongest factor in algorithm personalization, with

watching certain videos for a long amount of time and liking posts following in terms of influence (Boeker and Urman 2022). In terms of the level of perceived influence, a study identified that the follow-feature has the strongest role. It also discussed the context of the formation of filter bubbles on TikTok and the proliferation of problematic content in this regard (Boeker and Urman 2022; Holone 2016), with different aspects underlined for content creators (Klug et al. 2021). The exposure is limited to narrow perspectives related to any content that they may follow. This may lead to more prone cognitive biases such as confirmation, availability heuristics, or bandwagon effect, which may become a topic of concern by itself in the context of problematic use. At the same time, users can shape their own experiences by not following accounts or by providing feedback by pushing the button "Not interested" (Scalvini 2020). Here is a key point of questioning: if cognitive biases and curated feeds may lead indirectly to the shaping of personal values and identity, and if so, how can we analyze this?

### 4.3.2. TikTok Algorithm May Adjust Self-Beliefs and Self-Perceived Identity

We found that popular media has reported on user experiences related to thinking about the self, speculating that the videos shown to users say something about the type of person they are or even allow users to better understand their own social, sexual, and cultural identity and want to change it (Tiffany 2021; MacGowan 2020; Boseley 2021; French 2018). At the same time, changes in algorithms that conflict with goals spark emotional outrage, dissatisfaction, or platform exodus (DeVito 2021). But until now, no reference has been made through studies to reveal the deeper layers of one's identity and personal values.

### 5. Discussion and Conclusions

We found a few studies, most of which were qualitative semi-structured interviews or focus groups, exploratory, with an average of 20–40 participants, that used online or live interview techniques on the general consumption and experience of TikTok and other social media platforms, highlighting similar concepts and models of thinking related to self-perceived identity and personal values that aligned with what our review has intended to study. Also, most of the participants included in the studies were teenagers and young adults, content consumers, which may represent a concern considering that content creators have a different perspective.

Most of the studies agree that the influence of the algorithm acts as a fluid and dynamic gatekeeper for users accessing or sharing information about virtually any topic. While the algorithm can contribute through Personal Engagement to create a filter bubble effect, it does not have direct knowledge of an individual's values or consciously aim to shape them. There is a need for developing alternative methodologies, testing correlations, performing quantitative studies, and further developing theoretical and methodological synergies. Even if it is ultimately up to users to critically evaluate and reflect on the content they consume and make informed decisions about their personal values and self-perceived identity, more structural knowledge and critical thinking programs may help users at being mindful of their media consumption habits.

Of course, academic analysis of social media algorithm ontology needs to examine and interpret users not as isolated entities but rather as dynamic and fluid nodes in a network who show and hide valuable data at the same time, like a crystal temporarily lightened and darkened in a more extensive social and cultural but opaque ecosystem that waits to be discovered. All the elements that comprise these digital spaces influence one another and co-evolve, changing the identity, personal values, and ultimately the behaviors of the users. Our aim is to deepen our understanding of each node influencing another and the whole network itself, as the possible consequences of the data invisibility or filter bubbles on radicalization, hyper-individualism, and their derived concepts of cult-like movements, post-humanitarian sensibility, or the digital savior complex, among others, may be unexpected.

**Author Contributions:** C.G.I. and M.L. contributed to the conception, structure of the paper, contributed to analysis, and interpretation of available literature. C.G.I. contributed to the development of the initial draft. M.L. reviewed and critiqued the output for important intellectual content. Both authors contributed to the article equally and approved the sbmitted version. All authors have read and agreed to the published version of the manuscript.

**Funding:** This research received no external funding.

**Informed Consent Statement:** Not applicable.

**Conflicts of Interest:** The authors declare no conflict of interest.

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
