# Peer review of "Are TikTok Algorithms Influencing Users’ Self-Perceived Identities and Personal Values? A Mini Review"

_socsci, doi:10.3390/socsci12080465_

Round 1

Reviewer 1 Report

This is a solid and very compact review, and I am happy to recommend it for publication.

  Just a small number of minor suggestions:   1. In the abstract, the authors state that TikTok poses "special challenges for researchers" and say that they "discuss further topics of interest related to the subject and possible perspectives as well as recommendations regarding future research in this area."  Why not be more specific and give some indication of what the special challenges are and at least hint at the recommendations that will be given?   2. In the introduction, the authors state: "personalized content bought by the algorithm, can enhance positive experiences and promote psychological wellbeing while in the same time can lead through cognitive biases to the „echo-chamber” effect to isolation, rigid policy attitudes or radicalization in various aspects of personal values and beliefs with major potential societal impact not yet fully addressed."  Not disagreeing with these claims, but could use some citations, more so because outcomes we take as self-evident don’t always hold, e.g., one study found looking at opposing views on Twitter increases commitment to one’s old views.   3. In the intro, the authors describe TikTok as the "most advanced and sophisticated algorithm system."  I'm always wary of claims about "the most," "least," etc. since they are hard to definitively prove and also typically temporary.  Why not just say that TikTok has an extremely advanced and sophisticated algorithm system.   4. From the cognitive sciences, I'm familiar with the term "folk psychology," but not clear on what it adds to say "folk theories" as opposed to just "theories"   5. Concluding paragraph in the intro is vague.   6. There's an errant scare quote in the title for section 3.   7. The two sentences in the following quote seem contradictory: "Yet, little is known about how literate the users are. It is shown that the 83 degree of algorithmic literacy impacts the overall experience and influences the social media personal experience."  The authors state little is known and then give a case of something significant being known.   8. Not clear on why "Personal Engagement," "Identity Strainer," "Algorithmic privilege” and "Algorithmic representational harm" are capitalized. Nor do I understand the varied ways in which they are capitalized.  Is this because of how the developers of the terms deploy them?    9. "Dynamic" does not fit so well with the crystal metaphor.   10. You might briefly want to connect the networked self to ideas about socio-technologically extended self from cognitive science.  There's no shortage of materials here, but a few papers worth considering are Crippen's (2023) "Emotional Environments" and Crippen and Rolla's "Faces and Situational Agency," both published in Topoi.  You might also go to classic literature, e.g.,  Clark and Chalmers' "The Extended Mind" (published in Analysis), though this deals more with technologically extended minds than socially extended one.

English mostly good, but see "Comments and Suggestions for Authors."

Reviewer 2 Report

I enjoyed reading the article - it will be of use to students and practitioners. Explicit definitions of the concepts mentioned in section 4.1 (e.g. 'self-identification', self-perceived identity and 'cognitive bias') would strengthen the article.

Some issues with grammar - see the list below.

Lines 

6: ...related to the TikTok algorithm 

12:  ...over an individual's sense of self-understanding 

29: A reference is required to support the statements. For example, does this relate to Smith and Short (2022)? If so, make it clear.

30: ...perspective of a values-aligned... 

35-39: Source? Does this refer to Bucher (2020)? 

36: Little academic analysis .... This sentence can do with revision. Examples of some of the small analyses? 

51: Comma after 'how individuals perceive them, 

114: which means... 

218: The TikTok algorithm .... of a user's personal values.... 

239: At the same time.
